# Generating a Precision Endoxifen Prediction Algorithm to Advance Personalized Tamoxifen Treatment in Patients with Breast Cancer

**DOI:** 10.3390/jpm11030201

**Published:** 2021-03-13

**Authors:** Thomas Helland, Sarah Alsomairy, Chenchia Lin, Håvard Søiland, Gunnar Mellgren, Daniel Louis Hertz

**Affiliations:** 1Department of Clinical Pharmacy, University of Michigan College of Pharmacy, Ann Arbor, MI 48109, USA; salsmry@med.umich.edu (S.A.); linchenc@med.umich.edu (C.L.); DLHertz@med.umich.edu (D.L.H.); 2Hormone Laboratory, Department of Medical Biochemistry and Pharmacology, Haukeland University Hospital, 5021 Bergen, Norway; gunnar.mellgren@uib.no; 3Department of Clinical Science, University of Bergen, 5007 Bergen, Norway; hsoiland@gmail.com

**Keywords:** endoxifen, 4OHtam, tamoxifen metabolism, CYP2D6, pharmacogenetics, SULT, UGT, CYP3A, CYP2C, personalized treatment

## Abstract

Tamoxifen is an endocrine treatment for hormone receptor positive breast cancer. The effectiveness of tamoxifen may be compromised in patients with metabolic resistance, who have insufficient metabolic generation of the active metabolites endoxifen and 4-hydroxy-tamoxifen. This has been challenging to validate due to the lack of measured metabolite concentrations in tamoxifen clinical trials. *CYP2D6* activity is the primary determinant of endoxifen concentration. Inconclusive results from studies investigating whether *CYP2D6* genotype is associated with tamoxifen efficacy may be due to the imprecision in using *CYP2D6* genotype as a surrogate of endoxifen concentration without incorporating the influence of other genetic and clinical variables. This review summarizes the evidence that active metabolite concentrations determine tamoxifen efficacy. We then introduce a novel approach to validate this relationship by generating a precision endoxifen prediction algorithm and comprehensively review the factors that must be incorporated into the algorithm, including genetics of *CYP2D6* and other pharmacogenes. A precision endoxifen algorithm could be used to validate metabolic resistance in existing tamoxifen clinical trial cohorts and could then be used to select personalized tamoxifen doses to ensure all patients achieve adequate endoxifen concentrations and maximum benefit from tamoxifen treatment.

## 1. Introduction and Objective

Tamoxifen is an endocrine drug that has significantly improved outcomes in hormone receptor positive breast cancer [1]. However, the 10-year recurrence rate in patients using tamoxifen is ~30% [1]. Tamoxifen is a weak anti-estrogen that requires metabolic conversion to active metabolites that have higher affinity to the estrogen receptor (ER). This conversion is mediated by several cytochrome P450 (CYP) enzymes in which CYP2D6 plays a central role in the generation of the main active metabolite endoxifen [2]. Diminished *CYP2D6* phenotypic activity is associated with lower levels of endoxifen [2,3] that may reduce treatment response [4], which we refer to as metabolic resistance [4,5,6]. However, few large clinical trials have measured endoxifen concentration, preventing confirmation of metabolic resistance or identification of an endoxifen threshold below which efficacy is compromised. Many studies have investigated the association of tamoxifen treatment outcomes with *CYP2D6* genotype, as a surrogate of endoxifen concentration (Figure 1). The inconclusive findings [7] from these analyses may be due to the imprecision in predicting endoxifen from only *CYP2D6* genotype [8].

This review comprehensively reviews the factors that contribute to imprecision in endoxifen prediction by *CYP2D6* and describes the contribution of other non-*CYP2D6* genetic and clinical factors to endoxifen concentrations. The objective of this review is to provide a better understanding of these factors to move this field away from the use of a single genotype (*CYP2D6*) and toward building a precision endoxifen prediction model that integrates genetic and clinical variables. This algorithm could enable definitive analyses of the association of endoxifen concentration with tamoxifen treatment effectiveness and could then be used for personalized tamoxifen dosing to ensure that all patients achieve target endoxifen concentrations and receive maximum benefit from tamoxifen treatment.

## 2. Tamoxifen Clinical Use, Metabolism and Metabolic Resistance

### 2.1. Tamoxifen Treatment and Mechanism of Action

Tamoxifen is a selective estrogen receptor modulator used in the treatment and prevention of hormone receptor positive breast cancer. In the adjuvant setting, tamoxifen (20 mg/day) is mainly indicated for premenopausal patients with a recommended treatment time of 5 years that may also be extended to 10 years [1,9]. Although the overall reduction in mortality from tamoxifen treatment is substantial, the recurrence rate reduction remains about 50% during treatment and one-third in the subsequent five years [1], with some patients experiencing treatment failure. Tamoxifen may also be used to reduce the risk of developing breast cancer in high-risk women with or without neoplastic events [10,11].

The rationale for tamoxifen treatment of hormone receptor positive breast cancer is to prevent transcription of ER-regulated genes involved in breast cancer differentiation, proliferation, and migration [12]. Tamoxifen’s mechanism of action involves binding the ligand binding pocket of the ER that promotes a conformational change preventing recruitment of coactivators and instead cause association with corepressors thus repressing transcriptional activity of the ER [13,14]. This process takes place because tamoxifen has a higher affinity to the ER than does estradiol (E2), its natural ligand.

### 2.2. Tamoxifen Metabolism

Tamoxifen has a highly lipophilic structure and has been shown to be 98% albumin bound, which contributes to its long half-life of around 7 days [15]. Steady state levels are reached within 4–8 weeks [16]. Upon oral administration, some first pass metabolism occurs in the small intestine [15] but the majority of tamoxifen metabolism occurs in the liver and follows the classic two phases. Phase I hepatic metabolism of tamoxifen involves the addition or removal of certain functional groups through N-oxidation, hydroxylation or demethylation catalyzed by various members of the cytochrome P450 (CYP) family of enzymes [17] (Figure 2). The metabolites formed directly from tamoxifen are described as primary metabolites. The major route of phase I tamoxifen metabolism is demethylation of tamoxifen to N-desmethyl-tamoxifen (NDtam) catalyzed primarily by CYP3A4/5, in addition to CYP2C19/9, CYP1A2 and CYP2D6 [17,18]. NDtam is the major metabolite of tamoxifen and its plasma concentrations are approximately double that of the parent tamoxifen at steady state [3,19]. An alternative and minor route involves hydroxylation of tamoxifen to the active metabolite 4-hydroxy-tamoxifen (4OHtam) catalyzed by CYP2D6, CYP2C9/19, CYP2B6 and CYP3A4 [20]. Other primary metabolites include 3-hydroxy-tamoxifen (3-OHtam), tamoxifen-N-oxide (tam-NoX) and α-hydroxy-tamoxifen [17,21].

The main active metabolite of tamoxifen, 4-hydroxy-N-desmethyl-tamoxifen, hereby and earlier referred to as endoxifen, is a secondary metabolite formed by hydroxylation of NDtam catalyzed by CYP2D6 alone and by demethylation of 4OHtam to endoxifen by CYP3A4/5, CYP2C19 and CYP2D6 [17]. Secondary metabolites such as N-didesmethyl-tamoxifen (N,N-DDtam), 3-hydroxy-N-desmethyl-tamoxifen (3OHNDtam) and N,N-didesmethyl-4-hydroxy-tamoxifen (norendoxifen) are also formed from the primary or secondary metabolites [22], however their affinity and ER-inhibition are minor or yet to be determined (Table 1).

The phase I metabolites are further conjugated by glucuronidation [21,23] or sulfation [24] in hepatic phase II metabolism reactions catalyzed by uridine glucuronosyltransferases (UGTs) and sulfotransferases (SULTs), respectively [25]. Conjugation deactivates metabolites and increases their water solubility for excretion via bile or urine, however as much as 69% of the metabolites in the bile are reabsorbed by enterohepatic recirculation [26,27].

**Figure 2 jpm-11-00201-f002:**
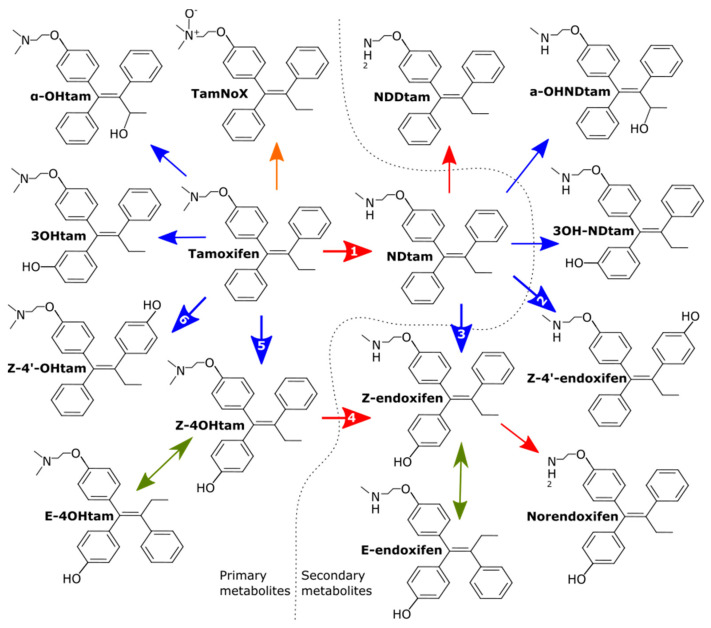
Overview of tamoxifen metabolites formed during first phase hepatic metabolism. Only isomers of 4OHtam and endoxifen are shown, other metabolites may also be subject to isomerism. Blue, red, orange, and green lines represent hydroxylation, demethylation, N-oxidation and non-enzymatic isomerization, respectively. Numbered arrows represent the following CYP450 enzymes involved in generation of active metabolites and their less active isomers: (1) CYP3A4/5, CYP1A1/2, CYP2C9/19, CYP2D6 (2) Unknown (3) CYP2D6 (4) CYP3A4/5, CYP2C19, CYP2D6. (5) CYP2C9/19, CYP2D6, CYP3A4, CYP2B6 (6) CYP2B6, CYP2D6. Modified from [28].

### 2.3. Active Metabolites: 4OHtam and Endoxifen

Tamoxifen is regarded as a prodrug as several of its metabolites have higher ER affinity and these metabolites may be responsible for the overall anti-estrogenic effect. Two of these metabolites (4OHtam and endoxifen) have up to 100× higher affinity to the ER compared to the parent drug tamoxifen [29,30] and are referred to as active metabolites (Table 1). Both active metabolites (4OHtam and endoxifen) may be formed as different isomers (E and Z isomers), in which the Z-isomers have >99% higher affinity to the ER compared to the E-isomers [31]. Adding to the complexity, the position of the hydroxyl group on the rings of hydroxylated metabolites may also vary. Z-endoxifen or Z-4OHtam with the hydroxyl group on the secondary structure, are referred to as 4-′-isomers or 4-prime-isomers (i.e., Z-4′-endoxifen and Z-4′-OHtam). The Z-4′-isomers exhibit ~10% of the binding affinity to the ER compared to the Z-isomers [32].

The more potent anti-estrogenic action of endoxifen and 4OHtam compared to tamoxifen itself is caused by the presence and position of the phenolic hydroxyl in their structures. In short, the positioning of the phenol group ensures deeper penetration into the binding pocket of the ER causing conformational changes in the ligand binding domain and correct positioning of the bulky side chain for antagonistic conformation of the ER [13,33]. The two active metabolites have similar affinity to the ER and similar anti-estrogenic effect on breast cancer cell proliferation [34]. However, since endoxifen plasma levels exceed 4OHtam by 5–10× it is regarded as the most important active metabolite. In contrast to 4OHtam, endoxifen has been shown to induce proteasomal degradation of the ER [35]. The same in vitro study also indicated that endoxifen induces highly concentration-dependent effects in blocking of ERα transcriptional activity and inhibition of estrogen-induced breast cancer cell proliferation, even in the presence of physiologically relevant concentrations of tamoxifen, NDtam, and 4OHtam [35]. Tumor growth inhibition in mouse models treated with endoxifen have also been shown to be highly concentration dependent [36]. Although other metabolites such as the 3-hydroxylated metabolites have been reported to have anti-estrogenic activity, their concentrations in vivo are generally too low to demonstrate any noteworthy antagonistic effect on the ER and to this point no other highly active metabolites have been described [28].

**Table 1 jpm-11-00201-t001:** Relative ER affinity and abundance of tamoxifen metabolites.

Metabolite	Relative ER Affinity	Relative Abundance
Tamoxifen	1	1
Tam-NoX	n.d	0.07
3OHtam	10	<0.01
Norendoxifen	n.d	
α-OHtam	n.d	<0.01
NDtam	0.85	1.78–2.01
NNDDtam	0.46	0.29–0.22
3OHNDtam	n.d	<0.01
**4OHtam Isoforms**		
Z-4OHtam	100	0.01–0.02
Z-4′-Ohtam	10	0.02–0.03
E-4OHtam	<0.03	<0.01
**Endoxifen Isoforms**		
Z-endoxifen	100	0.07–0.09
Z-4′-endoxifen	10	0.05–0.08
E-endoxifen	<0.03	<0.01

Affinity and abundance are relative to tamoxifen. Data on ER affinity was obtained from [31,32,34,37,38]. Information on abundance was extracted from studies using highly specific and selective LC-MS/MS assays [28,39]. n.d = not determined.

### 2.4. Associations of Active Metabolite Concentrations with Tamoxifen Efficacy

In addition to the pharmacological and pre-clinical evidence suggesting that concentrations of active metabolites are associated to the anti-estrogenic effect of tamoxifen, there is also evidence from clinical studies suggesting that active metabolite concentrations determine tamoxifen efficacy [4,5,6,40]. Three separate retrospective studies of patients treated with adjuvant tamoxifen have identified an association between low serum levels of endoxifen and inferior breast cancer outcome [4,5,40]. Two of the studies [4,5] identified similar thresholds of 16 nM and 14.15 nM, while the third study reported a lower threshold of 9 nM endoxifen [40]. Another small explorative study of 48 Asian women with breast cancer demonstrated a J-shaped relationship in which patients with endoxifen levels <53.6 nM and >187.4 nM were at higher risk of recurrence [41].

The first prospective study designed specifically to investigate metabolic tamoxifen resistance [42] found no association between relapse free survival and endoxifen levels as a continuous variable, quartiles or after applying the 16 nM threshold in 667 patients with early-stage breast cancer. However, heavy censoring due to patients switching to Aromatase Inhibitors (AIs) may have reduced the power of the survival analyses [43,44]. Metabolic resistance has also been investigated in the metastatic setting in two recent prospective studies. In 247 patients with metastatic breast cancer receiving 20 mg tamoxifen as first- or second-line treatment, no association was found between endoxifen concentration and overall response rate, progression free survival, or clinical benefit [45]. A secondary analysis of a prospective, randomized phase II trial investigating the clinical benefit of tamoxifen dose escalation in patients expected to have low endoxifen concentrations due to their *CYP2D6* genotype, as explained later, found similar endoxifen concentrations in patients who had and had not progressed after 6 months [46].

The relationship between endoxifen concentrations and outcome has also been explored in the preventive setting in high-risk women receiving low-dose tamoxifen, however no association between endoxifen concentrations and neoplastic events was identified in 97 premenopausal patients treated with 5 mg tamoxifen for 2 years [47]. A recent randomized placebo-controlled trial of low dose tamoxifen (5 mg/day) in the preventive setting [11] showed similar prophylactic effect of 5 mg tamoxifen as what has previously been reported for 20 mg, indicating that low endoxifen concentrations may still be effective in the preventive setting.

It has also been reported that patients with 4OHtam <3.26 nM have inferior breast cancer specific survival [40] and the threshold was recently confirmed in an independent patient cohort [6]. A simulation study found that inclusion of 4OHtam into an antiestrogenic activity score was predictive of survival [37], but not superior to using endoxifen levels alone [4].

Due to the heterogeneity in the results described above, it is still not fully established that active metabolite levels are predictive of tamoxifen efficacy. However, metabolic resistance is likely most relevant in the adjuvant setting as the most consistent results were reported from studies analyzing patients treated adjuvantly with tamoxifen alone for 5 years. All three retrospectively determined endoxifen thresholds [4,5,40] and the identification [40] and validation [6] of the 4OHtam threshold were conducted in the adjuvant setting. In addition, all these studies had long-term follow up, allowing for inclusion of late recurrences (>5 years) which make up the majority of the relapses in ER positive breast cancer [48]. Inability to prospectively validate metabolic resistance in the adjuvant setting [42,49] may be due to a preponderance of early relapses (<3 years) that are frequently attributable to proliferative factors such as high Ki67 [50], and to the contribution of endocrine resistance from somatic ER gene (*ESR1)* mutations [51], epigenetics [52] or loss of ER [53].

A well-powered adjuvant study with long-term follow-up that excludes recurrence events caused by somatic ER mutations or loss of ER may be necessary to validate insufficient metabolite concentrations as a biomarker for ineffective tamoxifen therapy. Since such a study may never again be attempted, an alternative approach is necessary. One such potential approach may be to use the currently available studies with measured endoxifen concentrations to build a precision endoxifen algorithm that integrates clinical and genetic variables to predict steady state endoxifen concentrations in individual patients (Figure 3). That algorithm could then be used to retrospectively test whether predicted endoxifen concentration is associated with recurrence or survival in prospective tamoxifen clinical trials with long-term follow-up that do not have measured endoxifen concentrations including the Breast International Group 1-98 (BIG1-98) [54] and Arimidex, Tamoxifen, Alone or in Combination (ATAC) trials [55]. Upon validation of the association and determination of the target endoxifen concentration threshold, this algorithm could be used to select an appropriate starting dose of tamoxifen to ensure all patients achieve therapeutic endoxifen concentrations and receive maximal tamoxifen treatment benefit.

## 3. Prediction of Active Tamoxifen Metabolite Concentrations

### 3.1. Genotype-Predicted Endoxifen Concentrations and Tamoxifen Treatment Efficacy

As most studies do not have endoxifen concentrations available, an alternative approach has been to use genetic variations in the pharmacogene *CYP2D6* as a surrogate of endoxifen concentrations to investigate the endoxifen-efficacy association. Over 30 studies have applied this approach and have produced heterogenous results, extensively reviewed elsewhere [7], which have led to conflicting clinical recommendations on the utility of *CYP2D6* genotyping to inform tamoxifen dosing [7,56,57,58]. The inconsistent *CYP2D6*-efficacy results may be due to the imprecision in estimating endoxifen. In the following sections we will review factors that improve endoxifen prediction accuracy, including comprehensive *CYP2D6* genotyping, standardized phenotype translation, and awareness of tamoxifen adherence and concomitant *CYP2D6* inhibitor-use. Endoxifen prediction could be further enhanced by accounting for the contribution of other pharmacogenes involved in tamoxifen metabolism including *CYP2Cs, CYP3As, UGTs* and *SULTs,* which have been investigated for association with tamoxifen treatment outcomes with similarly heterogenous results [8,55,59,60]. The following sections will review the studies that have reported associations with endoxifen concentration and highlight the factors that need to be considered when developing a precision endoxifen prediction algorithm.

### 3.2. Effect of CYP2D6 on Endoxifen Formation

CYP2D6 is the central enzyme in the metabolic bioactivation of tamoxifen by catalyzing the addition of a hydroxyl group to the phenyl ring of NDtam thereby forming the active metabolite endoxifen [23]. As the demethylation of tamoxifen to NDtam is the major route of tamoxifen metabolism (~90%) the quantitative contribution of CYP2D6 in endoxifen formation is significantly higher than the alternative minor route of endoxifen formation through demethylation of 4OHtam which involves several enzymes [18]. For this reason, CYP2D6 metabolic activity is the major determinant of endoxifen generation.

#### 3.2.1. CYP2D6 Genetic Variation and Activity Phenotype Translation

The *CYP2D6* gene is located on chromosome 22q13.1 and is highly polymorphic with over 100 allelic variants/haplotypes known [61]. These germline variations include Single Nucleotide Variants (SNVs), insertions or deletions of single or multiple nucleotides, copy number variations or other structural variations that may result in splicing defects, frameshifts, whole gene deletions or changes in gene expression. This may lead to abnormal protein expression or function and consequently impact CYP2D6 metabolic activity, and thereby affect the formation of endoxifen.

*CYP2D6* allelic haplotypes are classified according to the star allele nomenclature [62]. Each star allele has a designated functional activity and activity score provided by PharmVar [62] which is used to determine the metabolizer phenotype by combining the functional activity of haplotypes into diplotypes. In the star allele system, an allelic haplotype with pharmacogenomic relevance is classified by the gene name followed by a star and a number (e.g., *CYP2D6*4*). The allelic haplotype usually refers to a group of variants that tend to be inherited together, in which one or more of the genetic variants affects protein activity. Several additional variants that are not known to influence activity may also be part of the star allele and may define subvariants (e.g., *CYP2D6*2B*). Importantly, the core alleles will be part of all sub alleles and ensures that all sub alleles can be combined into a single a star number with functional relevance, whereas the non-consequential alleles may be shared across star alleles and should not be used to define star alleles. There are two notable limitations of interpreting studies that use the star allele system, or any other pharmacogenetic system. First, the star allele is not substrate specific and the values for the metabolic activity are not determined exclusively based on data from tamoxifen metabolism studies. Recent consensus *CYP2D6* guidelines highlight the potential to enhance accuracy in scoring metabolic activity using a quantitative, substrate-specific percentage activity system [63]. Second, studies test for a defined list of *CYP2D6* genetic variations and, by default, classify all non-carriers as wild-type (*CYP2D6**1) when they may in fact carry another functionally consequential variant that was untested. Prior pharmacogenetic studies have used a variety of approaches for translating *CYP2D6* genotype into metabolic activity phenotype [64] but recently two major pharmacogenetic consortiums and a panel of experts agreed to a consensus scoring system [63] (Table 2).

#### 3.2.2. Effect of CYP2D6 Genetic Variation on Endoxifen Concentrations

The association between *CYP2D6* genotype and endoxifen concentrations was first described by Stearns and colleagues [2] in a pilot clinical trial in 2002. In this small study (*n* = 12), patients with *CYP2D6* non-functional variants (*4, *6, *8) had significantly lower levels of endoxifen compared to patients not harboring these variants (wild type). Two subsequent studies [3,65] reported 30% and 74% reductions in endoxifen concentrations for patients homozygous for non-functional alleles (*3, *4, *5, *6) compared to wild type. Taken together, these three early studies showed that patients using tamoxifen carrying *CYP2D6* non-functional alleles have significantly reduced levels of endoxifen. To date, more than 30 studies have investigated this association and *CYP2D6* genotype is now an established predictor of endoxifen concentrations (Table 3). These studies have found that 8% to 46% of endoxifen concentration variability can be explained by CYP2D6 genotype (regression R squared values). The higher endoxifen estimates are likely representative of the actual contribution of *CYP2D6*, while the lower estimates are possibly caused by a variety of factors including inconsistencies in inclusion criteria, genotyping comprehensiveness, translation of genotype to phenotype, adjustment for other clinical variables, and differences in the endoxifen measurement assays. The following section will review six factors that may have influenced the CYP2D6-endoxifen relationship, which also likely contribute to the inconsistent results from studies investigating the association of *CYP2D6* genotype or endoxifen concentration with tamoxifen treatment efficacy [66]. 

#### 3.2.3. CYP2D6 Genotyping Comprehensiveness

Insufficient comprehensiveness in determining *CYP2D6* * alleles and CNVs contribute to inaccurate prediction of CYP2D6 metabolic activity as demonstrated in a study re-genotyping 492 breast cancer patients using Roche’s CYP450 AmpliChip assay that demonstrated significantly improved stratification of patients into activity groups and reduced misclassifications [85]. Endoxifen concentrations were not measured in the population, but the results indicate that more comprehensive genotyping would allow for greater detection of the pharmacogenetic association. The first studies to report on the *CYP2D6*-endoxifen association interrogated only a handful of no-function alleles thereby misclassifying patients as wild-type/normal metabolizers. Only one of these studies [3] reported an r^2^ value (23%). Out of the 30 studies presented in Table 3, only 6 studies were genotyped using the Roche’s CYP450 Amplichip assay which interrogates 33 distinct *CYP2D6 * *alleles and is considered the gold standard in *CYP2D6* genotyping. Surprisingly, the two studies that reported r^2^s and were genotyped with Amplichip were among the bottom 5 studies in terms of r^2^ estimates. There was no clear trend in terms of genotyping comprehensiveness and r^2^ estimates. However, most of the studies (21/30) had determined 7 or more different CYP2D6 * alleles, which covers the most common CYP2D6 variants including intermediate function alleles. Most studies (19/30) also determined copy number variations (CNVs) and were thus able to identify ultra-rapid metabolizers which increased the accuracy of genotype to phenotypic activity prediction [86]. Determination of CNVs may be of equal importance as detection of rare alleles by the AmpliChip assay.

#### 3.2.4. CYP2D6 Activity Phenotype Scoring

Use of different systems to translate *CYP2D6* genotype to phenotype has led to inconsistencies in metabolic activity phenotype assignments between laboratories and investigators and introduces an additional challenge in comparing results between studies [87]. Prior to the implementation of the consensus scoring system [63], the functional activity diplotypes (i.e., IM/EM) were frequently used as the phenotype or further combined into metabolic activity phenotype groups (i.e., UM, EM, IM, PM). Alternatively, some studies used the Activity Score (AS) system [88], which assigns each allele an activity score and sums the allele scores into one activity score that may be used a continuous variable or categorized into metabolizer groups.

At least three different scoring methods (metabolizer groups, diplotypes and the activity score) were utilized in the 30 studies presented in Table 3 and the scoring methods used was relatively evenly distributed between the studies (11, 8 and 9 respectively). Of the 11 studies that reported an r^2^ estimate six utilized diplotypes to score CYP2D6 activity and reported unadjusted r^2^s between 15% and 38%. Among these studies there were also variations within the same scoring methods, such as recoding UM alleles to NM alleles [72]. The previous version of the AS score [88] was utilized by two studies, one of which [5] reported r^2^ estimates of 33–38% within 3 ethnic subgroups. The second study [73] did not report an r^2^ estimate for endoxifen but reported that 30% of the variability in the metabolic ratio between endoxifen and NDtam was explained by *CYP2D6* genotype. It should be noted that analyses of metabolic ratio usually produce higher r^2^ estimates as it more directly reflects the conversion of NDtam to endoxifen. The three studies that categorized patients into metabolizer groups reported unadjusted r^2^ estimates of 10%, 18%, and 26%, though that final estimate was adjusted for clinical variables including age and race.

#### 3.2.5. Racial Differences in CYP2D6 Allele Frequencies

Another factor that may impair the prediction of endoxifen by *CYP2D6* is not accounting for the differences in frequencies of *CYP2D6* genetic variations among racial groups [62]. For example, the reduced activity *CYP2D6**10 allele is carried by ~40% of Asian individuals but <6% of Africans and Europeans. Conversely, *CYP2D6**17 is found in ~22% of African Americans and <1% of Asians and Europeans [62]. It is therefore reasonable to assume that more comprehensive genotyping is of greater importance in cohorts with large racial diversity. Similar to most pharmacogenetics research, these studies primarily included Caucasian patients (19/30) with the remaining studies including, six Asian cohort, three mixed cohorts and a Native American/Alaskan cohort. The Asian study reported the lowest r^2^ among the cohorts that provided endoxifen estimates. As mentioned earlier, *CYP2D6**10 is common in Asian patients with an allele frequency of 52% and has been shown to reduce endoxifen levels in Asian patients to a similar degree as non-functional alleles [74,89,90]. The recent consensus system updated the scoring of *CYP2D6**10 from 0.5 to 0.25 [63], which may have improved the predictive performance of CYP2D6 genetics [87]. An analysis of 20 breast cancer cohorts using tamoxifen indicated that rescoring the *CYP2D6**10 allele to an activity of 0.25 increased the prediction of endoxifen in Asian cohorts and to a lesser extent in Caucasians [91]. The study by Saladores et al. reported separate r^2^s for the different ethnic groups in which the Asian cohort accrued in Singapore produced the largest r^2^ estimate. The lowest estimate (33%) was found in the cohort collected in the UK, which might indicate greater racial diversity.

### 3.3. Concomitant Use of CYP2D6 Inhibitors

Unlike the external sources of endoxifen variability such as genotyping comprehensiveness and adherence, CYP2D6 inhibitor use is a legitimate source of biological variability as concomitant drugs reduce CYP2D6 activity via phenoconversion [2,3]. Inclusion of patients using potent CYP2D6 inhibitors increases misclassification and thereby reduces accuracy of endoxifen prediction. Studies investigating the *CYP2D6*-predicted endoxifen-outcome association should therefore always account for potent CYP2D6 inhibitor use. This may be less important for the use of weak CYP2D6 inhibitors as they have not been consistently shown to significantly impact endoxifen levels as recently reviewed elsewhere [8].

A reasonable assumption would be that removal of strong inhibitor-users would increase the predictive power of CYP2D6. Inhibitor users were excluded from 11 of the 30 studies and from 3 of the studies reporting r^2^ estimates in Table 3 and these studies were randomly distributed across the table ranked according to R square estimates. However, several datasets used inhibitor data as an adjustment variable, which contributed to increased prediction of endoxifen levels together with CYP2D6. Due to current guidelines for these well-established drug interactions [50], patients recruited to recent studies are less likely to have been co-prescribed strong CYP2D6 inhibitors, though a large cohort analysis indicates the efficacy reduction from CYP2D6 inhibitors may be limited [92].

### 3.4. Tamoxifen Adherence

The results in Table 3 originate from studies that include patients at steady state and due to tamoxifen’s long half-life there is little variation expected to occur throughout the day and in terms of timing of dosing. However, a major external source of endoxifen variability is adherence to the drug [93,94,95] which reduce the association between CYP2D6 and endoxifen concentrations and is thus an obvious factor that should be accounted for when associating *CYP2D6* to clinical outcomes. The 30 studies summarized in Table 3 have applied distinct approaches for excluding non-adherent patients including the more classic approaches of self-reported compliance and pill-counts to using various pharmacokinetic cut offs based on tamoxifen concentrations. The tamoxifen cut offs ranged from excluding patients under the lower limit of tamoxifen quantification [70], removing patients with tamoxifen levels less than 10% of the mean [68] or by identification and removal of patients with deviating tamoxifen levels based on splits in the tamoxifen concentration distribution [5]. In addition, some studies used tamoxifen concentrations as an adjustment variable along with CYP2D6 and/or other factors to predict endoxifen. This approach may be questionable as tamoxifen concentrations are linearly related to endoxifen, thereby inflating the estimates [3]. All these approaches sought to identify and remove patients with poor tamoxifen adherence. The three studies with relatively strict pharmacokinetic adherence cut offs (10% of the mean/150 nM tamoxifen) all reported high r^2^s ranging from 27% to 38%. However, a study in an independent cohort found moderate correlation between the 150 nM adherence cut off and patient-reported non-adherence [93]. This indicates that low tamoxifen concentrations may also be pharmacology-related, perhaps via reduced absorption or increased first-pass metabolism, and that removal of such patients may artificially inflate the estimated contribution of CYP2D6 activity to endoxifen generation.

### 3.5. Endoxifen Measurement

Endoxifen may exist as different isomers (Z, E, Z-4-prime) [28] for which Z-endoxifen is the main active metabolite generated by CYP2D6. Selective LC-MS/MS methods are necessary to differentiate these isomers and avoid inaccurate Z-endoxifen measurement [96] which may obscure identification or verification of endoxifen predictors that may be important contributors in a possible prediction algorithm. Approximately half of the datasets (16/30) in Table 3 measured Z-endoxifen while the remaining studies did not separate the Z, Z-4-prime and E isomers. Of note, four of the top five studies in terms of explained endoxifen variability were all performed using z-isomer assays, while the study with the lowest explained variability was performed using total-endoxifen. This excludes the study that only reported an estimate based on metabolic ratio [73]. In addition, there were discrepancies among the studies in terms of using log formatted endoxifen concentrations in the regression analyses when obtaining the R squared values. Of note, most of the studies producing the highest r^2^ estimates used log or square root transformed endoxifen concentrations whereas most of the studies that did not transform the data had lower estimates.

### 3.6. Other Clinical Variables Associated with Endoxifen Concentrations

Several non-pharmacogenetic patient factors have been associated with steady state endoxifen concentrations including age, BMI, adherence, gender, and season and addition of clinical factors to multivariable regression models have been reported to predict endoxifen concentrations more accurately than by using *CYP2D6* as a single predictor [67,72]. These factors and their associations to tamoxifen and metabolite concentrations have recently been extensively reviewed elsewhere [8]. Briefly, increasing age has been positively associated with endoxifen and 4OHtam concentrations [97,98]. An inverse relationship is seen with BMI, in which high BMI is predictive of lower tamoxifen and metabolite levels [72,97]. There are also reports suggesting lower endoxifen concentrations during winter compared to other seasons [67,72], However, there are conflicting data regarding this putative association [8]. Clinical predictors of endoxifen concentration have been discovered in small patient cohorts, increasing risk of false positives and negatives. Therefore, definitive investigation of the contribution of these factors should be re-analyzed in a pooled analysis of datasets that collected variables that have been reported to contribute to variability in endoxifen concentrations.

### 3.7. Effect of CYP2D6 Genetic Variation on Other Active Metabolites

4OHtam has similar ER affinity as endoxifen and its concentrations may also contribute to tamoxifen efficacy, as previously reviewed. In addition to its role in endoxifen formation, CYP2D6 is involved in the metabolic conversion of tamoxifen to 4OHtam (Figure 2). *CYP2D6* genotype has been estimated to explain approximately 9% of the variability in 4OHtam concentrations [28] and 27.6% of the variability in the metabolic ratio (tamoxifen/4OHtam) [28]. There are multiple reports of increasing 4OHtam concentrations as *CYP2D6* metabolizer activity phenotype increases [68,73,89,99,100], as well as increasing activity scores [84]. Studies focusing solely on the diminished activity *CYP2D6**10 allele commonly found among Asians have reported lower 4OHtam concentrations for homozygous carriers of this variant compared to wild type patients [74,101]. Similar to endoxifen, the explanation of variability from *CYP2D6* is even greater when analyzing the metabolic ratio (tamoxifen/4OHtam) [70,80,84]. Although *CYP2D6* genotype is an important factor in determining 4OHtam concentrations, this association appears to be somewhat weaker than the relationship with endoxifen [32,41,82,102] and likely has a minimal contribution to the putative CYP2D6-efficacy relationship.

## 4. Contribution of Other Enzymes to Tamoxifen Metabolite Concentrations

*CYP2D6* metabolic activity explains at most approximately 50% of the variability in endoxifen concentrations. Validating the endoxifen-outcome association will likely require a precise endoxifen prediction algorithm that includes other variables that contribute to endoxifen concentrations. As illustrated in Figure 2, there are many enzymes involved in tamoxifen metabolism including CYP2Cs, CYP3As, SULTs, and UGTs. The following sections will summarize the reported associations for genetic variants in these enzymes with concentrations of tamoxifen, endoxifen, and other metabolites. While some of these associations have been consistently found across studies, as described below, some have only been detected in a single study and require further replication. Several studies have investigated associations for functionally consequential variants within other genes not described below, including CYP2B, flavin mono-oxygenase (FMO), and the P-glycoprotein/ABCB1 transporter; however, there is a lack of evidence of association for variation within these genes with concentrations of tamoxifen or of its metabolites [28].

### 4.1. CYP2Cs

CYP2Cs, including CYP2C8, CYP2C9, and CYP2C19, contribute to the metabolism of many xenobiotic compounds [103]. CYP2Cs contribute to the conversion of tamoxifen to Z-4-OHtam and CYP2C19 is also involved in the conversion of Z-4-OHtam to endoxifen, as illustrated in Figure 2 [28,72,104]. Compared with CYP2D6, the contribution of CYP2Cs to endoxifen generation is relatively small, explaining less than 3% [5].

*CYP2C9**2 (rs1799853; MAF = 0.119) and *CYP2C9**3 (rs1057910; MAF = 0.069) are common, diminished activity *CYP2C9* variant alleles. In human liver microsomes metabolism of tamoxifen to 4OHtam is decreased in *CYP2C9**2 and *CYP2C9**3 samples by 48% and 89%, respectively [105], which is similar to the effect of incubation with a CYP2C9 inhibitor [105]. Multiple studies have found that carriers of these reduced activity alleles have lower concentrations of active tamoxifen metabolites including endoxifen and 4OHtam (Table 4) [28].

*CYP2C19* polymorphisms have also been investigated for their effect on tamoxifen metabolism. *CYP2C19* has two common null activity alleles, *CYP2C19**2 (rs4244285; MAF = 0.148) and *CYP2C19**3 (rs4986893; MAF = 0.012) [106] and one common enhanced activity allele *CYP2C19**17 (rs12248560; MAF = 0.225) [99]. Clinical studies have reported that patients with higher *CYP2C19* metabolic activity phenotype have higher conversion of tamoxifen to its active metabolites, particularly 4OHtam [73,99].

CYP2C8 is another polymorphic enzyme within this CYP enzyme family. Out of the fourteen identified *CYP2C8* variant alleles, *CYP2C8**2 (rs11572103, MAF = 0.012), *CYP2C8**3 (rs11572080; MAF = 0.105), and *CYP2C8**4 (rs1058930; MAF = 0.04) are relatively common diminished activity variants [107,108]. *CYP2C8**2 is the most common in African American population, whereas *CYP2C8**3 is the most common in Caucasian populations [107,108]. One study reported univariate associations of *CYP2C8* activity phenotype with endoxifen; however, this association was attributed to linkage between *CYP2C8* and *CYP2C9* [72].

### 4.2. CYP3As

CYP3A, including CYP3A4 and CYP3A5, are highly active in drug metabolism [103]. CYP3A4 and CYP3A5 participate in the conversions of tamoxifen to NDtam and of 4OHtam to endoxifen [67]. Similar to *CYP2D6*, *CYP3A4* has several known polymorphisms that affect enzyme activity. One of the most well-established functionally consequential variants is *CYP3A4**22 (intron 6 C > T; rs35599367) [109], which reduces *CYP3A4* gene expression and enzyme activity approximately 2-fold in in vitro studies of human liver samples [110]. In clinical studies, plasma concentrations of endoxifen in *CYP3A4**22 carriers were up to 1.4-fold greater than wildtype patients, and carriers also had higher concentrations of tamoxifen, NDtam, and 4OHtam [67]. This increase in endoxifen concentrations is unexpected for a reduced activity polymorphism. It is hypothesized that patients carrying *CYP3A4**22 have higher levels of tamoxifen and all of its metabolites due to reduced intestinal CYP3A activity and first-pass metabolism, which would increase tamoxifen bioavailability [67]. Other studies have not reported a difference in concentrations of tamoxifen or its metabolites in *CYP3A4**22 carriers [71,84,111,112]

Other *CYP3A4* variants, such as *CYP3A4**1B (rs2740574; A > G) and *CYP3A4**18 (rs28371759; T > C), have not been extensively studied in tamoxifen bioactivation, partially due to inadequate allele frequency [73,113]. Moreover, *CYP3A4**20 (rs67666821), *CYP3A4**6, *CYP3A4**26 alleles have previously been identified as loss-of-function alleles but their polymorphic effects in tamoxifen bioconversion and treatment effectiveness have not been examined [114].

The contribution of genetic variants in CYP3A5 to endoxifen has also been studied. *CYP3A5**3 (rs776746) is a loss of function allele that is more common in African Americans (MAF = 0.13) than Caucasians (MAF = 0.01) [115]. Clinical studies have not reported any effect of *CYP3A5**3 on endoxifen levels [84,90,112,113,115,116,117], even in large studies that conducted subgroup analyses stratified by CYP2D6 and CYP2C9/19 phenotype [73].

### 4.3. SULTs

Sulfotransferases (SULTs) are phase II liver enzymes that catalyze sulfation of endogenous and exogenous compounds, thereby increasing the water solubility for excretion. There are 12 main SULTS, with SULT1A1 and SULT1A2 the most commonly expressed and studied isoforms [118]. SULT1A1 metabolizes the tamoxifen metabolites 4OHtam and endoxifen into 4OHtam sulfate and endoxifen sulfate, respectively [119]. *SULT1A1* is a highly polymorphic gene among different ethnicities [120]. Carriers of two variants in *SULT1A1* (rs6839, rs1042157) have been reported to have 10–15% higher levels of endoxifen and 4OHtam. Similarly, the most common variants of *SULT1A2* are *SULT1A2**2 (rsID27742, MAF = 0.00273), which is found in LD with *SULT1A1**2 [121], and *SULT1A2**3 (rs762632, MAF = 0.104) [119]. Patients carrying these *SULT1A2* variants also have higher endoxifen and 4OHtam metabolite levels [82]. However, in the same study *SULT1A1* genotypes did not demonstrate any associations between tamoxifen and its metabolites.

### 4.4. UGTs

UDP-glucuronosyltransferase (UGTs) are another group of phase II metabolic enzymes that prepare compounds for elimination by adding a glucuronide group. UGTs have been demonstrated to be involved in the phase II metabolism of tamoxifen and its metabolites. There are several commonly studied polymorphic UGT enzymes including UGT1A4, UGT2B7, and UGT2B15, and their role in the metabolism of tamoxifen has been previously reviewed in detail [122,123,124]. UGT1A4 contributes to the N-glucuronidation of tamoxifen and 4OHtam [122]. Multiple studies have reported that patients carrying the *UGT1A4* Leu48Val variant have higher concentrations of tamoxifen-N-glucuronide [28,123] and lower concentrations of other glucuronidated metabolites [125]. UGT2B7 is responsible for glucuronidation of endoxifen. *UGT2B7* is highly polymorphic, and *UGT2B7**2 (His268Tyr, rs24648760) has reduced in vitro glucuronidation activity [125]. Patients carrying *UGT2B7**2 have been found to have higher concentrations of tamoxifen and endoxifen, and lower concentrations of glucuronidated metabolites [81,125]. Finally, Romero Lorca et al. reported that patients carrying *UGT2B15* Lys523Thr or UGT2B17 deletion have higher concentrations of certain glucuronidated metabolites [125], though these unexpected findings require independent validation.

**Table 4 jpm-11-00201-t004:** Non-CYP2D6 Enzymes associated with tamoxifen, endoxifen, or other metabolites.

Gene	Variants	Metabolites Measured	Reported Association	Ref.
CYP2C9	*2 and *3	Tam, 4OHtam, endoxifen, glucuronides, E and Z	Lower 4OHtam and endoxifen	[28]
	*2 and *3	Tam, NDtam, 4OHtam, endoxifen, N,N-DDtam, norendoxifen	Lower 4OHtam/tam ratio	[5]
	*1–*11	Endoxifen	Lower endoxifen and end/4OHtam ratio	[73]
	*2 and *3	Endoxifen	Lower endoxifen	[72]
CYP2C19	*2	Tamoxifen, NDtam, tamNoX, Z-4′-OH-Tam, Z-4OH-tam, Z-endoxifen, Z-4′-endoxifen	Higher Tam/4OHtam ratio	[70]
	*2, *3, *17	Endoxifen	Higher 4OHtam	[73]
	*2, *3, *17	Tam, NDtam, 4OHtam, endoxifen, N,N-DDtam, norendoxifen	Lower norendoxifen/N,N-DD-tam ratio	[5]
	*2, *3, *17	Tam, 4OHtam, NDtam	Higher 4OHtam/tam ratio	[99]
CYP3A4	*22	Endoxifen	Higher endoxifen	[67]
	*22	Tamoxifen, NDtam, tamNoX, Z-4′-OHtam, Z-4-OH-tam, Z-endoxifen, Z-4′-endoxifen	Higher endoxifen	[70]
SULT1A1	rs6839, rs1042157	Tamoxifen, endoxifen, 4OHtam and NDtam	Higher endoxifen and 4OHtam	[119]
SULT1A2	*2 and *3	Tam, 4OHtam, NDtam, endoxifen, TamNoX	Higher endoxifen and 4OH-tam	[82]
UGT1A4	Leu48Val	Tam, 4OHtam, endoxifen, glucuronides, E and Z	Lower tam/Tam-N-glucoronide ratio	[28]
	Leu48Val	Glucuronidated metabolites of endoxifen and 4OHtam	Lower 4OHtam-N-Gluc and endoxifen-gluc	[125]
	Leu48Val	Tamoxifen, E and Z-endoxifen, E- and Z-4-OHtam and the corresponding glucuronides	Higher Tam-N-gluc	[123]
UGT2B7	*2 (His268Tyr)	Tam and endoxifen	Higher endoxifen	[81]
	*2 (His268Tyr)	Glucuronidated metabolites of endoxifen and 4OHtam	Higher Tam, 4OHtam, NDtam. Lower 4OHtam-O-gluc and 4-OH-Tam-N-gluc	[125]
UGT2B15	Lys523Thr	Glucuronidated metabolites of endoxifen and 4OHtam	Higher 4-OH-Tam-O-gluc and endoxifen-gluc	[125]
UGT2B17	Deletion	Glucuronidated metabolites of endoxifen and 4OHtam	Higher 4OHtam-N-gluc	[125]

## 5. Conclusions and Directions for Future Research

Building a precise endoxifen prediction algorithm requires integrating the genetic and clinical variables discussed above. Several approaches to building an integrated algorithm have been reported including using standard linear regression [67]. In recent years pharmacokinetic modeling approaches have been used to investigate the factors contributing to metabolite concentrations [94,97,126,127,128]. These studies generally identified the same important factors but provide more mechanistic understanding of their effects on endoxifen concentration variability [97,127,128] and could be more precisely adapted into personalized dosing algorithms to achieve target endoxifen concentrations [94]. Finally, novel machine learning techniques have been utilized to improve model precision, though translation of these models into clinical practice may be challenging [129].

The possibility that there are patients who receive diminished clinical benefit from tamoxifen treatment due to metabolic resistance has been heavily debated over the last two decades. Recent prospective trials investigating metabolic resistance have been negative both by using *CYP2D6* as a predictor of endoxifen steady state concentrations and by using endoxifen concentrations directly [42]. However, these prospective studies have premature follow-up data and or have been conducted in the non-adjuvant settings [45,46]. An alternative approach to validate the endoxifen-efficacy association would be to use existing data from large, prospective trials of adjuvant tamoxifen use to predict individual patient’s expected endoxifen concentration and test the association with long-term survival data. This strategy would be a vast improvement over previous attempts to use only *CYP2D6* genotype that have failed to validate the association [54,55]. In this review we have provided evidence-based recommendations on how best to conduct analyses to determine the contribution of CYP2D6 and other genetic and clinical variables to endoxifen concentration. Comprehensive *CYP2D6* genotyping, the use of updated standardized scoring of phenotypic activity and endoxifen assays that selectively measures Z-endoxifen are all factors that would allow for increased accuracy in endoxifen prediction. Awareness and removal of non-adherent patients will also allow for removal of external variability that is not attributed to CYP2D6 metabolic activity. The use of potent CYP2D6 inhibitors should also be accounted for to avoid inaccurate prediction caused by phenoconversion. We have also shown that other genetic and clinical factors contribute significantly to variability in endoxifen concentrations and the collection and adjustment for these variables increase the ability to predict endoxifen concentrations in patients with breast cancer.

To move this field forward and to validate the existence of metabolic resistance we firstly suggest generating a precision endoxifen prediction algorithm that properly accounts for all the factors that reduce the accuracy of predicting endoxifen from *CYP2D6* genotype and then to integrate other factors genetic and clinical factors that affect endoxifen concentrations. A precision endoxifen algorithm could then be used to re-analyze prospective tamoxifen trials that have long-term follow up data to validate the effect of metabolic resistance and identify the endoxifen threshold below which efficacy is compromised. If successful, this algorithm can further be used to select an appropriate tamoxifen starting dose to achieve target endoxifen concentrations to avoid under-treatment and improve treatment outcomes.

## Figures and Tables

**Figure 1 jpm-11-00201-f001:**
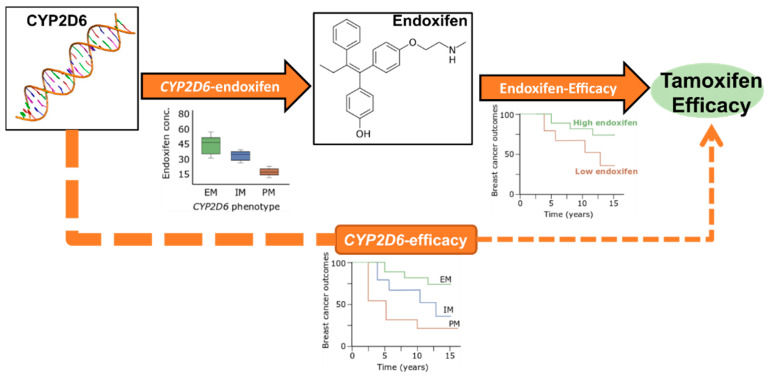
Approaches for investigating metabolic resistance in breast cancer patients. The *CYP2D6*-endoxifen relationship is well established. The endoxifen-efficacy relationship has been reported but not sufficiently validated due to a lack of available data. The *CYP2D6*-efficacy relationship has not been conclusively demonstrated due to the imprecision in using *CYP2D6* as a surrogate of endoxifen concentration. Box plot and survival curves are for illustration purposes only. EM = Extensive Metabolizer, IM = Intermediate Metabolizer, PM = Poor Metabolizer.

**Figure 3 jpm-11-00201-f003:**
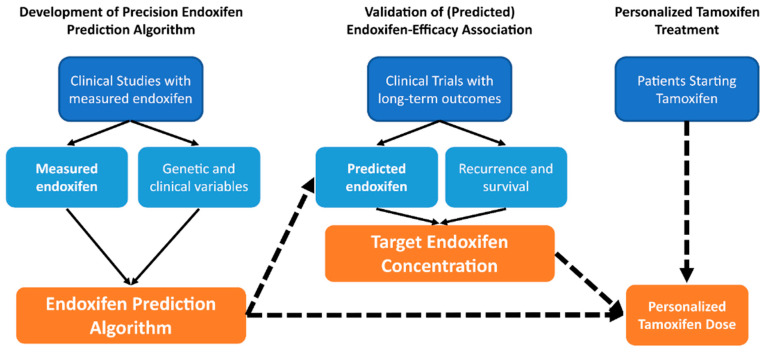
Approach for generating a precision endoxifen prediction algorithm to enable personalized tamoxifen dosing. A precision endoxifen prediction algorithm that integrates clinical and genetic variables could be generated from existing datasets. The algorithm would then be used to validate the association of endoxifen concentration with tamoxifen efficacy and identify the target endoxifen concentration. This information could then be used to inform personalized dosing that maximizes tamoxifen treatment efficacy.

**Table 2 jpm-11-00201-t002:** Activity Score System.

Allelic Activity Scoring			Metabolic Phenotype Scoring	
Activity	Genotype Examples	Activity Score	Activity Score Sum	Phenotype
High	*1XN, *2XN, *35XN	2	>2.25	UM
Normal	*1, *2, and *35	1	1.25–2	NM
Reduced	*9, *17, *29, *36, *41	0.5	0.25–1	IM
Slow	*10	0.25	0.25–1	IM
None	*3, *4, *5, *6, *7, *8	0	0	PM

UM = Ultra-rapid Metabolizer, NM = Normal Metabolizer, IM = Intermediate Metabolizer, PM = Poor Metabolizer.

**Table 3 jpm-11-00201-t003:** Overview of studies reporting *CYP2D6* genotypes in correlation with endoxifen concentrations.

Patients			CYP2D6 Genotype and Phenotype Scoring		Endox Assay	% Endox or Metabolic Ratio (MR) Explained r^2^, (Adjusted. r^2^)			Study
n	Race	Exclusion Criteria	CYP2D6 * Alleles	Scoring	Z/Total	Endox	MR	Covariates	Ref
236	Caucasian	Tam <10% mean	3, 4, 5, 6, 7, 9, 10, 41, CNV	DT	Sqrt-z	38.6	68	None	[28]
583	Mixed	Tam<150 nM, inhibitors	3, 4, 5, 6, 9, 10, 14, 15, 17, 41, CNV	AS	Log-z	33–38	53	Age, BMI, non-2D6 genetics	[5]
196	Caucasian	Unspecified tam cut-off	3, 4, 5, 9, 10, 41, CNV	DTs	Log-z	30.1 (46)	-	Age, BMI, genetics, inhibitors, season	[67]
279	Caucasian	Tam <10% mean	2, 3, 4, 5, 6, 7, 9, 10, 17, 41, CVN	DTs	Z	27	51	-	[68]
97	Caucasian	None	2, 3, 4, 5, 6, 7, 8, 9, 10, 11, 15, 17, 29, 35, 41, CNV	MG #	Total	(26)	(38)	Age, ethnicity	[69]
80	Caucasian	None	4, 6, 8	DTs	Total	23	-	None	[3]
1370	Caucasian	None	CYP450 AmpliChip, CNV	MG	Z	18 (46)	-	Age, BMI, race, tam conc	[4]
730	European	Tam<LLQ	4, 9, 10, 7, 6, 17, 41, CNV	DTs	Z	16.8 (19.4)	-	Inhibitors, non-2D6 genetics	[70]
178	Caucasian	Inhibitors, non-adherence	2, 3, 4, 5, 6, 7, 8, 9, 10, 11, 15, 17, 29, 35, 41, CNV	MG #	Z	(16)	-	CYP3A4, demographics	[71]
302	Caucasian	Inhibitors, non-adherence	CYP450 AmpliChip, CNV	DTs	Log-z	15.4 (23)	-	Weight, season, CYP2C9	[72]
224	Asian	Unspecified tam cut off, inhibitors	2, 3, 4, 5, 6, 10, 41	MG	Total	10	-	None	[41]
116	Caucasian	None	2, 2A, 3, 4, 5, 6, 7, 8, 9, 10, 11, 12, 14, 17, 29, 35, 41, CNV	AS	Total	-	29.6 (33.7)	Inhibitors	[73]
83	Caucasian	Endoxifen < LLQ	2, 2A, 3, 4, 5, 6, 7, 8, 9, 10, 12, 14, 17, 29, 41, CNV	MG	Z	-	-	None	[40]
202	Asian	Inhibitors or inducers	2, 5, 10, CNV	DTs *	Z	-	-	None	[74]
122	Caucasian	Tam <150 nM	2, 3, 4, 5, 6, 7, 8, 14, 9, 10, 18, 41, 14B, CNV	MG	Z			None	[75]
77	Caucasian	Inhibitors, non-adherence	2, 2A, 3, 4, 5, 6, 7, 8, 9, 10, 11, 12, 17, 41, CNV	AS	Z	-	-	None	[76]
98	Asian	SSRIs	5, 10, 21, 41	DTs	Z	-	-	None	[77]
117	Caucasian	Inhibitors	CYP450 AmpliChip, CNV	MG	Total	-	-	None	[78]
119	Mixed	None	2, 2A, 3, 4, 5, 6, 7, 8, 9, 10, 11, 12, 17, CNV	AS	Log-z	-	-	Inhibitors	[32]
35	Mixed	Inhibitors	2, 3, 4, 9, 10, 17, 29, 35, 41,	AS	Total	-	-	Race, menopausal status	[79]
667	Caucasian	None	CYP450 AmpliChip, CNV	MG *	Z	-	-	None	[42]
80	Asian	None	2, 5, 6, 10, 39, 41, CNV	AS	Total	-	-	Tam conc., BMI, non-2D6 genetics	[80]
59	Asian	Inhibitors, non-adherence	10	DTs	Total	-	-	None	[81]
152	Caucasian	Unspecified tam cut-off	2, 3, 4, 5, 6, CNV	MG *	Total	-	-	Age	[65]
120	Caucasian	None	CYP450 AmpliChip, CNV	DTs/MG	Total	-	-	Inhibitors	[82]
114	Caucasian	None	3, 4, 5, 6, 9, 41	AS/MG	Z	-	-	None	[83]
183	Asian	Inhibitors	2, 4, 6, 10, 14, 18, 21, 36, 41, 44, CNV	DT	Z	-	-	None	[46]
42	Native American	None	2, 3, 4, 5, 9, 10, 28, 33, 35, 41	AS/DTs	Total	-	-	Age, site, non-2D6 genetics	[84]
158	Caucasian	None	CYP450 AmpliChip, CNV	DTs	Total	-	-	Inhibitors	[19]

# metabolizer groups included EM-Fast and EM-Slow, * metabolizer groups included, hetEM. CYP450 AmpliChip interrogates the following CYP2D6*alleles: *1, *2, *4, *5, *10, *14, *16, *17, *22, *25, *29, *30, *33, *35, *36, *40, *41, *43, *45B, *46, *56B, *59, *64, *65, *73, *74, *84, *85, *86. AS = Activity Score, DT = Diplotypes, MG = Metabolizer Group, LLQ = lower limit of quantification, CNV = Copy Number Variation.

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
