# Peer review of "Generating a Precision Endoxifen Prediction Algorithm to Advance Personalized Tamoxifen Treatment in Patients with Breast Cancer"

_jpm, 2021, doi:10.3390/jpm11030201_

Round 1

Reviewer 1 Report

Generating a Precision Endoxifen Prediction Algorithm to Advance Personalized Tamoxifen Treatment in Patients with Breast Cancer titled paper is a work that addresses a topic of great interest.

I found some misspelled words:

Line 16. Inclonlusive - incorrect

Linr 45: theis -incorrect

Line 202: treamtent-incorrect

Page 12: clinal -incorrect

Sometimes abbreviations are used directly, without explanation, ex:

Line 13: 4-OHtam – when first time used

Figure 1: EM, IM, PM

Line 155: AIs

Line 182: ESR1

Table 2: UM, NM

Line 79: ER

One  sentence with error:

Line 54:   The relationship 54 CYP2D6-endoxifen relationship is well established

Regarding the content, I have only one remark to consider and include in discussions in the 3.4 chapter with the title Tamoxifen Adherence and 3.5 chapter entitled Endoxifene Measurement the J.M Nardin’s and collab article The Influences of Adherence to Tamoxifen and CYP2D6. Pharmacogenetics on Plasma Concentrations of the Active Metabolite (Z)-Endoxifen in Breast Cancer, Clin Transl Sci (2020) 13, 284–292; doi:10.1111/cts.12707

Author Response

Thank you for your detailed review. Please see the attached point-by-point response indicating the changes we made in response to each of your helpful suggestions. 

Reviewer 2 Report

The article is well prepared and presents the up-to-dae information in tamoxifen metabolism and genetcs. I have no major remarks. One small question is if you also searched for epigenetic regulation patterns in tamoxifen resistance e.g. as it was published in IJMS:

miRNA Expression Profiles in Luminal A Breast Cancer—Implications in Biology, Prognosis, and Prediction of Response to Hormonal Treatment Int. J. Mol. Sci. 202021(20), 7691; https://doi.org/10.3390/ijms21207691

Is there any difference in tamoxifen resistance / metabolism in breast cancer in young women? (younger than 40?)

Author Response

Thank you for your review. Please see the attached point-by-point response indicating the changes we made in response to each of your suggestions. 
